# How Do Flemish Laying Hen Farmers and Private Bird Keepers Comply with and Think about Measures to Control Avian Influenza?

**DOI:** 10.3390/vetsci11100475

**Published:** 2024-10-05

**Authors:** Femke Delanglez, Bart Ampe, Anneleen Watteyn, Liesbeth G. W. Van Damme, Frank A. M. Tuyttens

**Affiliations:** 1Faculty of Veterinary Medicine, Ghent University, 9820 Merelbeke, Belgium; femke.delanglez@ugent.be; 2Animal Sciences Unit, Flanders Research Institute for Agriculture, Fisheries and Food (ILVO), 9090 Melle, Belgium; bart.ampe@ilvo.vlaanderen.be (B.A.); anneleen.watteyn@ilvo.vlaanderen.be (A.W.); liesbeth.vandamme@ilvo.vlaanderen.be (L.G.W.V.D.)

**Keywords:** avian influenza, nets, confinement, poultry, self-evaluation

## Abstract

**Simple Summary:**

Avian influenza (AI) is an infectious disease and could lead to death, health problems, and economic losses. Therefore, this study aimed to gather information about the compliance with and perceived effectiveness of AI control measures (applicable during high-risk periods in Flanders, North of Belgium, in 2021) by professional laying hen farmers and private bird keepers. Overall, self-reported compliance was high among professional laying hen farmers but much less among private bird keepers. Among private bird keepers, compliance and perceived effectiveness were lowest for confining the birds indoors, whereas for farmers, it was lowest for placing nets over the free-range. This study highlights the need for information campaigns explaining to private bird keepers, particularly the need for the various AI control measures imposed. Should these campaigns prove unsuccessful, local authorities might need to implement stricter enforcement of existing control measures or explore alternative ways to increase compliance, such as information posters in relevant stores for private bird keepers or meeting private bird keeper interest groups to provide broader support.

**Abstract:**

Competent authorities of many countries, including Belgium, impose control measures (preventing wild bird access to feeders and water facilities, indoor confinement of captive birds, or fencing off outdoor ranges with nets) on professional and non-professional keepers of birds to prevent the spread of avian influenza (AI). Flemish laying hen farmers (FAR, n = 33) and private keepers of captive birds (PRI, n = 263) were surveyed about their opinion on and compliance with AI measures legally imposed during the most recent high-risk period before this survey in 2021. Participants answered questions on a 5-point Likert scale (1 = the worst, 3 = neutral, and 5 = the best). FAR indicated better compliance with the AI measures than PRI, except for net confinement. FAR indicated that they and other poultry farmers complied better with AI measures than PRI. Additionally, PRI indicated that they better complied than other PRI keepers. FAR regarded the AI measures as more effective than PRI. To prevent the spread of AI more effectively, national authorities could focus on information campaigns explaining to private bird keepers the need for the various control measures that they impose. If these campaigns fail, local authorities may need stricter enforcement or alternative ways to increase compliance.

## 1. Introduction

Avian influenza (AI) is a contagious viral disease to which nearly all poultry species are susceptible, including chickens, ducks, geese, turkeys, pheasants, guinea fowl, quail, and partridges, but also pigs, cattle, horses, dogs, cats, whales, foxes, sea lions, and even humans [1,2,3,4,5,6]. The most highly pathogenic subtypes of the AI virus cause severe disease symptoms (nasal and eye discharge, coughing, dyspnea, severe drowsiness, and diarrhea) and can be responsible for a mortality rate of up to 100% in infected poultry flocks [7,8,9,10]. Captive bird flocks can get contaminated with the AI virus through direct contact with contaminated wild birds, by consuming feed and water contaminated with the feces of wild birds, or by indirect transmission routes (e.g., clothes, equipment, vehicles) [11,12,13]. Especially wild migratory (water)birds are the reservoir of AI and seem to be the primary source of AI virus introduction in captive flocks [6,14,15]. Biosecurity measures are key to preventing the introduction and spread of disease agents that cause AI [12,16,17]. There are two categories of biosecurity, namely external (preventing the introduction of infectious agents from off-site into the farm) and internal (preventing the spread of infectious agents within the farm/flock), with specific measures for external biosecurity (reduction of (indirect) contact with infected wild birds, hygiene locks, wearing disinfected clothes and footwear) and for internal biosecurity (regular cleaning and disinfection, controlling movement of people and equipment within the farm) [12,18,19,20,21].

AI infection within poultry flocks results in significant economic losses and negatively impacts the welfare of the affected birds [3,22,23]. Prevention of AI infection is, therefore, a priority [24]. Furthermore, private birds have been involved in disease outbreaks such as Newcastle disease [25], Salmonella [26], and AI [27,28].The Belgian Government, like many other national authorities worldwide, therefore stipulates preventive measures during such high-risk periods, which (depending on the situation) apply to both non-professional (e.g., private bird keepers) and professional (e.g., farmers) bird keepers. These measures include indoor confinement of captive birds to avoid contact with wild birds, shielding off feed and water facilities from wild birds, prohibition of exhibitions, competitions, public markets of poultry and birds, establishment of safety zones in the perimeter of infected farms, and obligatory reporting of infected birds [29]. 

The effectiveness of measures to prevent the spread of AI, however, depends on how well they are implemented by bird keepers and farmers. It is crucial, therefore, to investigate how well bird keepers implement these measures in practice and whether they believe these measures are effective. This kind of information is often gathered by qualitative or quantitative surveys [30,31]. To illustrate compliance with biosecurity measures, Danish broiler farmers demonstrated a greater emphasis on compliance and the proper implementation of biosecurity measures during audit visits compared with periods without such inspections [32]. The need for an audit to better comply with biosecurity measures raises questions on the reliability of self-assessment by farmers on sensitive issues such as compliance with legally imposed norms. When people are asked to provide a self-assessment, the answer may be biased in a positive direction, and they may be motivated to give an answer that is most desired [33]. For example, in surveys in which a presumably representative sample of people is asked to self-evaluate socially valued traits (e.g., intelligence, attractiveness, and sociability), often more than half of the respondents believe to score above the average for such traits [34,35]. If the importance of social traits can push self-assessment in a desired direction, the same may be true for the self-reporting on compliance with imposed AI control measures. 

Furthermore, the best strategy for dealing with poor compliance with disease control measures may depend on whether or not the level of compliance is linked with the belief in the effectiveness of the various measures. If the responses are negative (low level of compliance), several solutions are possible, such as recommendations, penalties, and training of the farmer and personnel [36]. The attitudes and behaviors of the farmers, the farm staff, visitors, and private bird keepers play an essential role in the implementation of biosecurity and prevention of the spread and infection of AI, so the perceptions of the poultry sector and private bird keepers are of utmost importance [37,38,39]. Furthermore, confidence in the effectiveness of the measure on disease prevention can be influenced by social norms (opinions and actions of other people) [40,41], the self-efficacy (belief in the ability to execute a measure), and the motivation [42,43,44]. Additionally, there is often a gap between intentions and actual behavior [42], as well as between self-reported practices and what people do [45].

It is therefore important to understand the support and compliance of the current Belgian AI control measures by the bird keepers and farmers. In particular, there is little research on the role of private bird keepers in the spread of AI [46,47,48], which is an important knowledge gap to address. This study investigated, through an online survey, how effective the AI control measures imposed by the Belgian Government are perceived to be by both professional and non-professional bird keepers and how well they comply with those measures themselves. Because the sample of respondents cannot be claimed to be random and because the possibility cannot be excluded that respondents may give a socially desirable (instead of a truthful) answer, the participants were asked not only whether they complied but also whether their colleagues and other poultry keepers complied with AI control measures. In addition, specifically for the professional laying hen farmers, their opinion on the effectiveness of extra AI control measures imposed by the Belgian Government was explored, and that is why they had a more comprehensive questionnaire.

## 2. Materials and Methods

Two different groups of people were contacted to complete a survey on AI preventive measures: private keepers of poultry and/or other captive birds (PRI) and professional laying hen farmers (FAR). The survey was open from 31 July 2021 until 30 September 2021 (Appendix A).

### 2.1. Survey Procedure

An online survey link was distributed to PRI through social media (Facebook and LinkedIn), e-mail, and newsletters of Flemish bird associations (Vlaams Neerhof, Steunpunt Levend Erfgoed vzw, and Nationale Raad voor Dierenliefhebbers). Flemish FAR (n = 186) were contacted by e-mail obtained via the Department of Agriculture and Fisheries (Flanders). After two weeks, FAR were contacted by phone with the request to complete the survey if they had not already done so.

### 2.2. Survey

The survey was created and accessible with the LimeSurvey software version 4.4 (www.LimeSurvey.org (accessed on 31 July 2021)) and could be completed anonymously by participants. In the first part of the survey, both PRI and FAR were questioned on the type of poultry/birds they kept, housing conditions, and whether their birds had access to an outdoor range. Additionally, FAR were asked to give their age. In the second part of the survey, PRI and FAR were questioned on their level of compliance with four general nationwide AI control measures: (1) compulsory confinement of poultry or other captive birds to prevent contact with wild birds by keeping them indoors or by (2) covering outdoor ranges with nets (mesh size of max. 10 cm), (3) obligation to feed and water poultry and other captive birds indoors or by other means which prevent contact with wild birds, (4) a ban on feeding or providing water to poultry and other captive birds from surface water supplies or rainwater to which wild birds have access unless the water has been treated to inactivate viruses. These general measures had been imposed by the Belgian government during the most recent AI high-risk period before the survey [29], namely between 15 November 2020 and 6 April 2021 for PRI and between 1 November 2020 and 12 May 2021 for FAR. PRI and FAR were asked to evaluate how well they complied with the general measures themselves (‘self’) but also how well they thought their fellow bird keepers (‘colleagues’ of their respective group) and people from the other group (PRI or FAR group opposite of the respondent’s category) complied. In the third part of the survey, PRI and FAR were asked to give their opinion on the effectiveness of these four general AI measures. In the fourth and last part, FAR were questioned on additional biosecurity measures specific for professional laying hen farmers: (1) cleaning vehicle, (2) research disease, (3) prohibition access from risk area, (4) cleaning vehicle abroad, (5) prohibition trade of animals, (6) disinfection bath in front of stable, and (7) registration of visitors on the farm. They were asked to indicate how effective they considered these to be. The questions from Part Four were not asked at PRI. 

All questions of the survey, except for those in the first part, could be answered with a 5-point Likert scale. The scale ranged from 1 (no compliance/not effective) to 5 (full compliance/most effective). For each question, the option ‘not applicable’ was available as well.

### 2.3. Statistical Analysis

Response data were analyzed using R (version 4.2.1). Responses to each question with scores between 1 and 5 were analyzed with linear mixed models.

In questions about the indication of the level of compliance by PRI and FAR themselves and in comparison with their colleague FAR and PRI, the fixed factors were the respondent group (PRI or FAR), the AI control measure (net confinement, separation of feed and water from wild birds, indoor confinement, and prohibition using feed and water from resources accessible by wild birds), and their interaction. The respondent ID was included as a random factor to correct for repeated answers by each respondent. A similar model was used for the analysis of the comparison of the effectiveness by PRI and FAR of the AI control measures.

Questions for the indication of FAR about additional biosecurity measures, the answer option (cleaning vehicle, research disease, prohibition access from risk area, cleaning vehicle abroad, prohibition trade of animals, disinfection bath, registration of visitors) was the fixed factor, and the respondent ID was the random factor. 

If respondents chose the ‘not applicable’ option for any questions, this was considered to be a missing value and not taken into account in the statistical analysis. In tables, 95% confidence intervals were reported based on the least squares means. In cases of significant differences, a post hoc pairwise comparison with Tukey correction was used to correct the *p*-values for multiple testing. A value of *p* < 0.05 was considered significant. The data were assumed to be sufficiently normally distributed based on a graphical evaluation of the residuals of the used models (QQ-plot, histogram, and residuals versus fitted value plot).

## 3. Results

### 3.1. Response Rate and Respondent Demographics

The survey was filled out by 263 PRI and 33 FAR respondents. For FAR, this corresponds to a response rate of 18% of the contacted laying hen farmers, and 64% of them had a free-range area for their laying hens. The majority of FAR (83.5%) were aged between 35 and 65 years, whereas 14% were younger than 35 years, and 2.5% were older than 65 years. For PRI, neither the response rate nor their age distribution is known. Birds owned by PRI respondents were chickens (86%), pigeons (24%), water birds (24%), songbirds (8%), pheasants (7%), turkeys (6%) and other bird species (5%). Of the PRI respondents, 86% had a free-range area for their birds.

### 3.2. Self-Evaluated Compliance with General AI Control Measures

Overall, PRI, and especially FAR, self-reported to comply rather well with the nationwide general AI control measures themselves (Figure 1A,B). Mean compliance scores by PRI ranged between 3.2 ± 0.08 for ‘Indoor confinement’ and 4.2 ± 0.08 for ‘Separation of feed and water from wild birds’ (Figure 1A). Mean self-compliance scores for FAR ranged from 4.1 ± 0.21 for ‘Net confinement’ to 4.7 ± 0.20 for ‘Indoor confinement’ and ‘Separation of feed and water from wild birds’ (Figure 1B). These scores are all significantly higher than the self-compliance scores given by PRI, except for ‘Net confinement’ (‘Separation of feed and water from wild birds’ *p* = 0.04 and ‘Prohibition to use feed and water from resources accessible by wild birds’ *p* = 0.01). The difference in the level of self-compliance reported by PRI versus FAR was largest for ‘Indoor confinement’ (3.2 vs. 4.7, *p* < 0.0001). 

### 3.3. Compliance with General AI Control Measures by Others

PRI reported better compliance with each AI control themselves as compared with other private bird owners (all *p* < 0.05) (Figure 1A). When PRI were asked to evaluate also the level of compliance by FAR, these scores were significantly higher than the scores they had given to fellow private bird keepers (all *p* < 0.05). For one of the four general AI control measures (namely ‘Indoor confinement’), the compliance score given to FAR was higher than the self-compliance score given by PRI to themselves (*p* < 0.0001) (Figure 1A). In summary, PRI respondents did not seem to think other private bird owners had the same level of compliance as themselves or those of FAR. 

This negative opinion about the level of compliance by private bird owners was even more pronounced among the FAR respondents (Figure 1B). The mean compliance score FAR gave to PRI was below the neutral point of the scale (score = 3) for all four general AI control measures. These scores were also lower than the level of compliance FAR reported for themselves (all *p* < 0.0001) or for their fellow FAR (all *p* < 0.0001). Compliance level by FAR was lower for ‘Net confinement’ than for the three other general AI control measures.

### 3.4. Opinion on the Effectiveness of General and Specific AI Measures

All four general AI control measures were thought to be more effective by FAR than PRI (all *p* < 0.0004) (Figure 2). The difference was most pronounced for ‘Indoor confinement’, which was given the lowest effectiveness score by PRI (2.5 ± 0.08) but a very high score by FAR (4.5 ± 0.18) (*p* < 0.0001). ‘Separation of feed and water from wild birds’ was given the highest effectiveness score both by PRI (3.6 ± 0.08) and by FAR (4.7 ± 0.18) (Figure 2). The ‘Prohibition to use feed and water from resources accessible by wild birds’ was also given a relatively high score by both groups of respondents. ‘Net confinement’, however, was given a relatively low effectiveness score by both PRI (2.7 ± 0.08) and FAR (3.5 ± 0.18). 

FAR respondents were also asked to evaluate the effectiveness of seven additional biosecurity measures specific to the poultry industry. The mean scores for these additional measures ranged between 3.9 (‘Registration of visitors’) and 4.3 (‘Cleaning vehicle’) (Figure 3).

## 4. Discussion

This survey is unique in providing information about the level of compliance with, and the opinion on the effectiveness of, the measures against AI in Belgium by two key stakeholder groups, namely professional laying hen farmers (FAR) and private keepers of birds (PRI). The outcomes of this study are important as the success of AI control measures imposed by the authorities may greatly depend on the level of support and compliance by those who are expected to implement them.

The number of respondents was higher for PRI (n = 263) than for FAR (n = 33), despite extra efforts to recruit FAR. Persuading poultry farmers to participate proved challenging, and personal outreach was needed by contacting them via phone, resulting in more willingness to fill out the survey. As our FAR respondents account for only 9% of all laying hen farmers in Flanders (368 are registered laying hen farmers in Flanders in 2022) [49], representativeness cannot be assumed. Indeed, laying hen farmers with a free-range system seem to be over-represented, possibly because some of the AI control measures affect them most and because free-range birds are more susceptible to the spread of AI [50,51]. Moreover, our client and financier instructed us not to include poultry farmers other than laying hen farmers. In general, there was no control over who filled out the survey. This could cause bias (e.g., people who are more confident in their expertise are more likely to participate in such surveys) [48]. However, the survey for PRI was distributed over different platforms, such as social media (Facebook and LinkedIn), e-mail, and newsletters of Flemish bird associations. Furthermore, the survey was distributed online, so farmers or private bird keepers without internet access could not participate. Thus, the groups with internet access were over-represented which is a common bias in surveys [48,52]. This could also result in an age bias because older people are less willing to participate in online surveys [53].

Nevertheless, an (anonymous) online survey method was used to map out information about the level of compliance and perceived effectiveness of the AI control measures because it allows obtaining data relatively fast and inexpensively. Moreover, such data are more difficult to obtain by interviews, in which the risk of the interviewer influencing the responses is expected to be greater, especially if it concerns sensitive issues such as compliance with legislation [54]. Even in anonymous online surveys, though, the possibility cannot be excluded that the respondents’ answers are not entirely truthful, e.g., because respondents may lack an unbiased view about their actions and/or wish to make a favorable impression, i.e., socially desirable responding [55,56]. As this may lead to misreporting, and in particular about the level of non-compliance with biosecurity measures, which may be met with disapproval by peers and society at large, we asked the respondents not only to evaluate compliance by themselves but also by their colleagues or peers. Perceived level of compliance by fellow bird keepers may thus better reflect reality than self-reported level of compliance, as it reduces socially desirable reporting [44].

Intriguingly and in contrast with the PRI respondents, the level of self-compliance reported by FAR respondents did not differ with how well they thought fellow laying hen farmers complied with the general AI measures. As both respondent groups indicated that the level of compliance was higher among FAR than other PRI, this perception probably matches reality (although, of course, no deductions about the actual compliance level can be made based on the survey of this study). This could be explained by the fact that the financial and other consequences of an AI outbreak tend to be much bigger for FAR than PRI [22,24,29,57,58,59,60]. Consequently, awareness about and support for the control measures can be expected to be higher among FAR than PRI, as well as the checks of compliance by local authorities and peers. 

‘Indoor confinement’ stood out as a general AI control measure for which compliance among private bird keepers (contrary to laying hen farmers) appears very low. A possible explanation is that private bird keepers may be less likely to have suitable facilities to keep their birds indoors for a prolonged period as compared with professional farmers. Consequently, long-term indoor confinement can negatively impact the behavior and welfare of birds due to a diminishing amount of natural behavior that can be performed in indoor facilities [61]. This effect is possibly more pronounced for privately kept birds, as these birds are less likely to have access to suitable indoor facilities (e.g., appropriate size, designated areas). The resistance to keeping birds indoors may thus be expected to be greater among private bird keepers versus laying hen farmers. The lower compliance with ‘Indoor confinement’ by PRI compared with FAR could also be related to the fact that an AI outbreak can have dramatic economic consequences for the poultry farmer, such as the depopulation of the entire flock of thousands of birds and even financial ruin, which is often less of a concern by the private bird keepers [23,48,57,58,62,63]. Poultry farmers and private bird keepers are commonly located near each other, so transmission is possible and could also negatively affect private bird keepers [64,65]. Thus, the apparent lower level of compliance among PRI may be related to their lower awareness of the transmission possibilities and the importance of minimizing transmission to control AI. This low compliance could reduce the effectiveness of control measures, which in turn could lead to increased infection risk for birds kept by both PRI and FAR.

Placing nets over the free-range area or using a roof or canvas cover has also been suggested as a protective measure against AI infection in a poultry flock [66,67,68]. However, the level of compliance by laying hen farmers was the lowest for this measure and considerably lower than for the three other general AI control measures. As flock sizes on commercial farms are much larger than is the case for the modal private bird keeper, the total outdoor area that would need to be netted makes this option logistically and financially very hard [69]. For professional laying hen farmers, keeping the hens indoors is often a much easier option to comply with AI control measures. In addition to the level of compliance, the perceived effectiveness of placing nets over the free-range area was also investigated. PRI found placing nets over the free-range area less effective compared with FAR.

In general, both respondent groups indicated that FAR better or equally complied with all other general AI control measures (‘Indoor confinement’, ‘Prohibition on using feed and water from resources accessible by wild birds’, and ‘Separation of feed and water from wild birds’) compared with PRI. This could be linked to the fact that it is easier to check compliance among laying hen farmers than private bird keepers due to the registration of their farms and the higher concentration of animals at fewer locations. Private bird keepers need to be registered only if they have more than 199 birds not destined for the food chain or if they regularly buy or sell birds [70]. Thus, the non-registered private bird keepers are difficult to reach by the government, which potentially reduces their ability to control the spread of AI in private bird keepers [52]. In the UK, approximately 20% of the private bird keepers reported they had little knowledge about the legislation in the UK about AI measurements, due to the lack of communication between them and the authorities and a lack of understanding of the legal requirements [47,48,69,71]. Furthermore, there is a possibility that the majority of private bird keepers ignore basic biosecurity measures and the potential risk of zoonotic diseases to humans [72]. Therefore, strategies should be constructed to improve knowledge and awareness and provide accessible and reliable information to private bird keepers [48,73].

Especially when the level of supervision by peers or local authorities is low, the level of compliance with AI control measures can be expected to be linked to how effective they are perceived to be. It is not surprising, therefore, that for PRI there seemed to be a negative association between the level of compliance with the various AI control measures and how effective they believed these to be. Ensuring proper implementation of these AI measures is crucial to preventing the spread of AI. This could in part be achieved by informing the private bird keepers about the AI measures and by emphasizing the importance of compliance with these measures by reaching them through a campaign or by placing informative posters with clear guidelines in the stores where they buy their birds or where they buy food and accessories for their birds for broader support. Furthermore, an alternative way of increasing compliance with AI measures is organizing meetings with private bird keeper interest groups to define control measures and explain consequences for animal welfare and human health. Thus, an essential prerequisite for improving compliance would be for the authorities and other stakeholders to provide and disseminate accessible and reliable information about the prevention of the spread of AI and stress the consequences for animal and human health [48,65,73,74,75,76] to change private bird keepers’ beliefs on effectiveness.

## 5. Conclusions

In conclusion, based on this survey, it seems that compliance with AI measures applicable in Belgium was high among professional laying hen farmers but less so among private bird keepers. The level of compliance generally mirrored how effective the respondents considered these measures to be. Among private bird keepers, compliance and perceived effectiveness were lowest for keeping the birds indoors, whereas for farmers it was lowest for placing nets over the free-range, which could be linked to the feasibility of these control measures within these specific contexts. To combat the spread of AI more effectively, national authorities could focus on reliable and accessible information campaigns explaining to private bird keepers, in particular, the need for the various control measures that they impose. If such campaigns fail to have the desired effect, the local authorities may need to consider stricter enforcement by more efficient supervision of the control measures or to consider alternative ways to increase compliance (e.g., informative posters with clear guidelines in relevant stores for private bird keepers and meeting with private bird keeper interest groups to define control measures that are broadly supported). 

## Figures and Tables

**Figure 1 vetsci-11-00475-f001:**
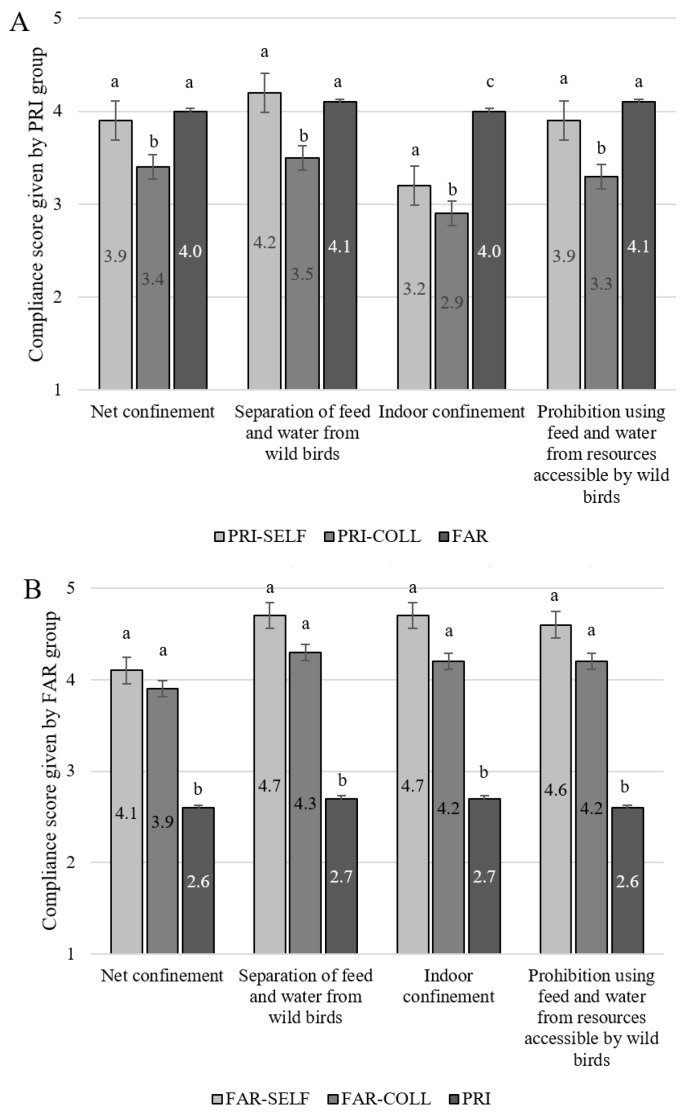
(**A**) Compliance scores on the general AI control measures given by the private bird keepers (PRI, N = 263) to themselves (PRI-SELF), towards colleagues (PRI-COLL), and towards laying hen farmers (FAR). (**B**) Compliance scores on the general AI control measures given by the FAR group (N = 33) to themselves (FAR-SELF), towards colleagues (FAR-COLL), and towards the PRI group. Scores range from 1 (no compliance) to 5 (most compliance). Numbers within bars represent the mean compliance score. Significant differences (*p* < 0.05) between bars in a specific measure are indicated with a,b,c scripts.

**Figure 2 vetsci-11-00475-f002:**
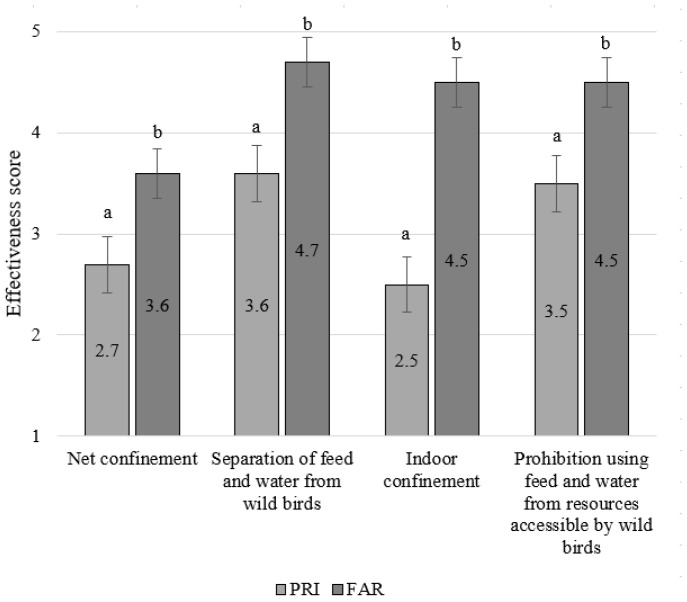
Comparison of the opinions of private bird keepers (PRI, n = 263) and laying hen farmers (FAR, n = 33) about the effectiveness of the general AI control measures. Scores range from 1 (not effective) to 5 (very effective). Numbers within bars represent the mean compliance score. Significant differences between FAR and PRI are indicated with a,b superscripts.

**Figure 3 vetsci-11-00475-f003:**
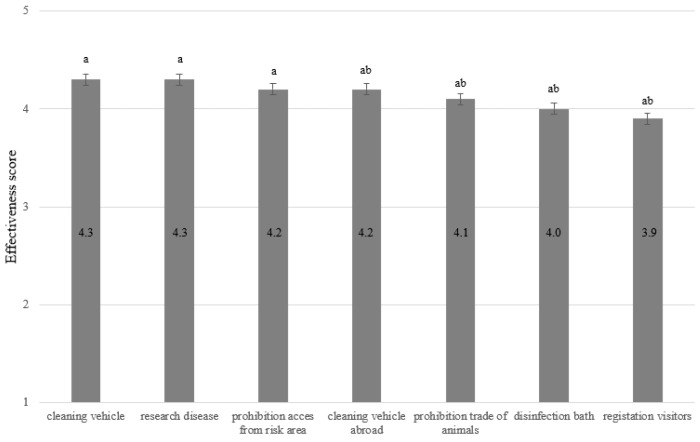
The opinion of laying hen farmers (n = 33) on the effectiveness of specific AI control measures. Scores range from 1 (not effective) to 5 (very effective). Numbers within bars represent the mean effectiveness score. Significant differences between bars are indicated with a,b superscripts.

## Data Availability

The original contributions presented in the study are included in the article material, further inquiries can be directed to the corresponding author.

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
