# Peer review of "How Do Flemish Laying Hen Farmers and Private Bird Keepers Comply with and Think about Measures to Control Avian Influenza?"

_vetsci, 2024, doi:10.3390/vetsci11100475_

Round 1

Reviewer 1 Report

Comments and Suggestions for Authors

General:

The topic is very relevant in the context of preventing spread of avian influenza. It is interesting to look into the perceptions and compliance of farmers and private bird keepers. Also, including not only self-evaluation but also peer-evaluation and evaluation of the other category of poultry owners is an interesting approach. However, the current manuscript has several major flaws that make it in my view not acceptable for publication unless the data is either analyzed or presented and discussed in a very different way. Because it would be too elaborate to point out all issues, I sum up some general issues and give some examples. This may help make improvements to the manuscript, after which potentially detailed review would become relevant but maybe not for this journal. That is up to the editor.

1) Data analysis is unclear: the description of the linear mixed models (and not providing a script or outputs of the models (estimates and 95% CI) that might resolve some unclarities) is completely unclear, including the random effects etc. Also the choices of fixed factors and interactions are unclear and how the final models looked like used for the results is not mentioned. The data itself (the whole survey) is also absent: would have been nice as supp file. In the figure captions it is not said which model was used. It says also 'that non-applicable answers were omitted, also from the analysis'  but its definition and describing this process is not written in mat and methods. 

2) Literature: a) The cited literature is very out of date, especially considering the changes with regard to avian influenza epidemiology, outbreak dynamics, biosecurity measures etc.; b) It also contains a lot of non-scientific (non-English) sources and simple handbooks for which better sources are available; c) Some references are incomplete, and cannot be found easily (like 7 and 8 and many more);

d) The literature is not well-focused on the topic of biosecurity in livestock owners or surveys with livestock owners.

- Numerous papers have been written about biosecurity in poultry owners (both farmers and backyard) with a survey included, compliance, perceptions etc., but these are not included. How about the outputs of the Netpoulsafe project (even in Flanders)??

- The lack of good literature also explains the focus on only the better-than-average effect, which is not so relevant in this context (more related to human psychological studies I guess), whereas other much more relevant biases in this context are ignored. These are described in papers with a similar topic, so I would suggest to read those first. I give some links below.

- It also explains flaws in reasoning. For example, what is mentioned briefly in the discussion (sensitivity of questions, but with an old, not-livestock related reference) is probably much different for poultry farmers (they have a lot more to loose) than for backyard poultry owners but this is not mentioned. Also, the problems that backyard owners have with indoor housing of water birds is very different from laying hen farmers that have a nice poultry house for which indoor housing of their chickens does not provide so much practical issues (and has also been described in similar studies in detail) is not discussed. Also the age bias (not many old farmers included, because they may not like online surveys) have been discussed in many papers, but not here.  And how much do the private bird keepers actually contribute to the spread/control of AI? Is there evidence of the role of backyard poultry for the spread in 2020-2021 or later? There is also literature on that.

In conclusion: If such papers would have been read, much more relevant types of biases and much more logical assumptions and conclusions on the obtained data could have been possible.

3) Some scientific reasoning and basic scientific paper structuring flaws: a) mentioning alternative measures in line 21 without explanation in the manuscript; b) words like remarkable, and even higher than and much more pronounced, and lower (although still high) in the results are subjective statements. There is also jumping to conclusions a lot (without support by the presented data): such as statements on differences which are not differences, like in 182-184: FAR respondents believed that the level of compliance by FAR colleagues was only slightly lower than the level of self-compliance although the differences were never statistically significant (all P > 0.05). Lines 206-209: The mean scores for these additional  measures ranged between 3.9 (‘Registration of visitors’) to 4.3 (‘Cleaning vehicle’) which are numerically lower than the FAR effectiveness scores for the general AI control measures except ‘Net confinement’ (compare Figures 2 and 3).

- Discussion:  a) the reasoning in lines 253-257 in in contradiction with itself. When people would have the better-than-average effect problem, how do you explain the higher evaluation of farmers by PRI? And why would peer-evaluation be more reliable than self-evaluation, considering these better than average effects? How fair is their evaluation then of other peers?; b) line 260-261: This strengthens the conclusion that the level of compliance by FAR is truly higher than by PRI. You did not measure actual compliance so I would not dare to say that; c) I have read three different conclusions: which is it? 302 in discussion: 'negative association between the level of compliance with the various AI control measures and how effective they believed these to be. Ensuring proper implementation of these AI measures is crucial to prevent the spread of AI. This could in part be achieved by informing the private bird keepers about the AI-measures and by emphasizing the importance of compliance with these measures by reaching them through a campaign or by placing informative posters with clear guidelines' ; 314 in conclusion: 'To combat the spread of AI more effectively national authorities could focus on information campaigns explaining to private bird keepers, in particular, the need for the various control measures that they impose' ; in abstract line 33: ' To prevent the spread of AI more effectively, national authorities could focus on information 33 campaigns explaining to private bird keepers, in particular, the need for the various control 34 measures that they impose, implement alternative control measures that have broader support or 35 implement stricter enforcement of the control measures'. Broader support in this conclusion, how? That is not discussed much. And do you want to inform them more or enforce more? It does not become clear from the different conclusions.

So in conclusion; The topic and data may be interesting but based on what is shown in the paper I cannot assess the validity of the methods/analysis, nor can I completely follow the discussion and conclusions. This is mostly because the data/analyses are not discussed in the context of relevant studies and also many statements contradict each other or suggest a certain subjectivity about the results, while lacking in this current form sufficiently sound scientific reasoning. 

And here some relevant papers:

McClaughlin E, Elliott S, Jewitt S, Smallman-Raynor M, Dunham S, Parnell T, Clark M, Tarlinton R. UK flockdown: A survey of smallscale poultry keepers and their understanding of governmental guidance on highly pathogenic avian influenza (HPAI). Prev Vet Med. 2024 Mar;224:106117. doi: 10.1016/j.prevetmed.2024.106117.

Jewitt S, Smallman-Raynor M, McClaughlin E, Clark M, Dunham S, Elliott S, Munro A, Parnell T, Tarlinton R. Exploring the responses of smallscale poultry keepers to avian influenza regulations and guidance in the United Kingdom, with recommendations for improved biosecurity messaging. Heliyon. 2023 Aug 17;9(9):e19211. doi: 10.1016/j.heliyon.2023.e19211.

Amalraj A, Van Meirhaeghe H, Lefort AC, Rousset N, Grillet J, Spaans A, Devesa A, Sevilla-Navarro S, Tilli G, Piccirillo A, Żbikowski A, Kovács L, Kovács-Weber M, Chantziaras I, Dewulf J. Factors Affecting Poultry Producers' Attitudes towards Biosecurity. Animals (Basel). 2024 May 29;14(11):1603. doi: 10.3390/ani14111603.

Tilli G, Laconi A, Galuppo F, Mughini-Gras L, Piccirillo A. Assessing Biosecurity Compliance in Poultry Farms: A Survey in a Densely Populated Poultry Area in North East Italy. Animals (Basel). 2022 May 30;12(11):1409. doi: 10.3390/ani12111409

Nespeca R, Vaillancourt JP, Morrow WE. Validation of a poultry biosecurity survey. Prev Vet Med. 1997 Jul;31(1-2):73-86. doi: 10.1016/s0167-5877(96)01122-1. 

Souillard R, Allain V, Dufay-Lefort AC, Rousset N, Amalraj A, Spaans A, Zbikowski A, Piccirillo A, Sevilla-Navarro S, Kovács L, Le Bouquin S. Biosecurity implementation on large-scale poultry farms in Europe: A qualitative interview study with farmers. Prev Vet Med. 2024 Mar;224:106119. doi: 10.1016/j.prevetmed.2024.106119.

Correia-Gomes C, Sparks N. Exploring the attitudes of backyard poultry keepers to health and biosecurity. Prev Vet Med. 2020 Jan;174:104812. doi: 10.1016/j.prevetmed.2019.104812.  

Laconi A, Tilli G, Galuppo F, Grilli G, Souillard R, Piccirillo A. Stakeholders' Perceptions of Biosecurity Implementation in Italian Poultry Farms. Animals (Basel). 2023 Oct 18;13(20):3246. doi: 10.3390/ani13203246.

Delpont M, Racicot M, Durivage A, Fornili L, Guerin JL, Vaillancourt JP, Paul MC. Determinants of biosecurity practices in French duck farms after a H5N8 Highly Pathogenic Avian Influenza epidemic: The effect of farmer knowledge, attitudes and personality traits. Transbound Emerg Dis. 2021 Jan;68(1):51-61. doi: 10.1111/tbed.13462. 

Delpont M, Salazar LG, Dewulf J, Zbikowski A, Szeleszczuk P, Dufay-Lefort AC, Rousset N, Spaans A, Amalraj A, Tilli G, Piccirillo A, Devesa A, Sevilla-Navarro S, van Meirhaege H, Kovács L, Jóźwiak ÁB, Guérin JL, Paul MC. Monitoring biosecurity in poultry production: an overview of databases reporting biosecurity compliance from seven European countries. Front Vet Sci. 2023 Aug 15;10:1231377. doi: 10.3389/fvets.2023.1231377. 

Comments on the Quality of English Language

Just a few examples of where English can be improved (only some, as I expect that the paper needs revising anyway because of the scientific content): a) infected poultry flocks cause ...(in line 55): poor English, as the infections are associated with these losses but not caused by the flocks, and you can also not say that the infected flocks are detrimental to the welfare of infected birds; b) lines 69-72 are unclear and also not in line with the next line on audits; c) Line 92: the participants were not only asked to report compliance by themselves but also by their peers (i.e. other bird keepers or poultry farmers). This seems incorrect phrasing. 

Author Response

See the attachment with the revision.

Reviewer 2 Report

Comments and Suggestions for Authors

The authors aim to gather information about the compliance with, and perceived effectiveness of, AI control measures by professional laying hen farmers and private bird keepers. The survey results are of great significance for effective prevention and control of AI. 

My only suggestion is to mark the source of the surveyed persons on the map, so that we can intuitively understand the geographical distribution of the survey results.

Author Response

Comment 1: The authors aim to gather information about the compliance with, and perceived effectiveness of, AI control measures by professional laying hen farmers and private bird keepers. The survey results are of great significance for effective prevention and control of AI. My only suggestion is to mark the source of the surveyed persons on the map, so that we can intuitively understand the geographical distribution of the survey results.

Response 1: Thank you for pointing this out. This sounds very interesting, however, the survey was anonymous, so it is impossible to know where the poultry farmers and private bird keepers are located.

Round 2

Reviewer 1 Report

Comments and Suggestions for Authors

The manuscript has been improved but especially in the introduction, but also elsewhere, it becomes clear that the actual concepts of biosecurity and avian influenza epidemiology are poorly understood. The references are rather good, but it looks like the content was not read or understood of these papers. This leads to statements that are not really correct. Not sure whether it is a combination of not being proficient in English or in biosecurity / epidemiology terminology or both, but I am afraid if people read it as such, they do not grasp the concepts as well. So although the survey part and analyses are much more clear and more discussion on potential biases are mentioned, these statements, that are too numerous for me to provide 'corrections' for (especially when it is not clear what you mean) regretfully force me to advice against publications. When you would consult a person that is experienced in writing on biosecurity/epidemiology and avian influenza, the paper may be submittable again but as such I do not recommend publication.

I added comments to the PDF to show the biggest issues. 

Comments on the Quality of English Language

See previous section

Author Response

See the attachment with the revision.

Round 3

Reviewer 1 Report

Comments and Suggestions for Authors

The manuscript was substantially improved. I am pleasantly surprised by how the paper has changed and think that is it now (almost) acceptable in the current form. Maybe the final minor remarks can be considered by the authors. In contrast with my prior remarks on the earlier versions, I now feel that most statements have been properly supported/more nuance is provided but two more statements could be improved:

- Lines 302-304: Moreover, the use of outdoor ranges (if these are provided) by laying hens on commercial farms can be rather poor, and often much lower as compared to the use of outdoor facilities by birds kept by private owners.

Is this a fact that can be supported by 'expert opinion' or literature or can it be stated with a bit more caution? And what is meant with 'the use can be rather poor'. Do you mean that chickens do not use it, or that farmers keep the locks to the outside range closed? That should be clarified here. And these matters depend completely on the type of range and farmer so I think it not easy to be generalized.

- Line 359-361: Thus, the government should provide accessible and reliable information about the prevention of the spread of AI, and stress the consequences for animal and human health [48,64,72-75] to change private bird keepers' beliefs on effectiveness.

I agree that this could be part of it, but it is a rather limited advice, compared to how much broader it is formulated in the conclusion. It suggests that only the government (and not the local authorities) are the only ones to come into action now by providing more information. Whereas, I think there are many ways to motivate private poultry owners and not only initiated by the 'government'. For example, in the Netherlands, the competent authority had a video online (FYI: https://www.nvwa.nl/onderwerpen/vogelgriep-preventie-en-bestrijding/ophokplicht#anker-4-afschermplicht-voor-dierentuinen-kinderboerderijen-en-houders-van-hobbyvogels) of how the birds can be kept away from wild birds, but this video was also mentioned/linked at websites for backyard poultry owners. Another option could be some kind of compensation for taking these measures (or free material to be collected somewhere to help change the housing) or a phone number people can call to get advice. I can also imagine that the poultry sector or universities and applied sciences schools could be involved in dissemination and providing context to the need for compliance by appearing in TV programs or at social media. I would certainly not mention all the potential options, just clarifying my point here, but it would be best to leave that last sentence of the discussion a bit more open. For example, in stead of saying that 'the government should' you can say that 'a very important prerequisite for improving compliance would be for the authorities and other stakeholders to provide......and disseminate......etc.'. That sounds less limiting but many other ways can be used to formulate this as well. 

Author Response

Thank you very much for your feedback, we are pleased to hear that the manuscript is substantially improved and that most statements have been properly supported.

Comment 1: 

The manuscript was substantially improved. I am pleasantly surprised by how the paper has changed and think that is it now (almost) acceptable in the current form. Maybe the final minor remarks can be considered by the authors. In contrast with my prior remarks on the earlier versions, I now feel that most statements have been properly supported/more nuance is provided but two more statements could be improved:

- Lines 302-304: Moreover, the use of outdoor ranges (if these are provided) by laying hens on commercial farms can be rather poor, and often much lower as compared to the use of outdoor facilities by birds kept by private owners.

Is this a fact that can be supported by 'expert opinion' or literature or can it be stated with a bit more caution? And what is meant with 'the use can be rather poor'. Do you mean that chickens do not use it, or that farmers keep the locks to the outside range closed? That should be clarified here. And these matters depend completely on the type of range and farmer so I think it not easy to be generalized.

Respons 1: Thank you for pointing this out. We agree with this comment. We have, accordingly, modified the text in Line 302 – 307 on page 8.

Comment 2: 

- Line 359-361: Thus, the government should provide accessible and reliable information about the prevention of the spread of AI, and stress the consequences for animal and human health [48,64,72-75] to change private bird keepers' beliefs on effectiveness.

I agree that this could be part of it, but it is a rather limited advice, compared to how much broader it is formulated in the conclusion. It suggests that only the government (and not the local authorities) are the only ones to come into action now by providing more information. Whereas, I think there are many ways to motivate private poultry owners and not only initiated by the 'government'. For example, in the Netherlands, the competent authority had a video online (FYI: https://www.nvwa.nl/onderwerpen/vogelgriep-preventie-en-bestrijding/ophokplicht#anker-4-afschermplicht-voor-dierentuinen-kinderboerderijen-en-houders-van-hobbyvogels) of how the birds can be kept away from wild birds, but this video was also mentioned/linked at websites for backyard poultry owners. Another option could be some kind of compensation for taking these measures (or free material to be collected somewhere to help change the housing) or a phone number people can call to get advice. I can also imagine that the poultry sector or universities and applied sciences schools could be involved in dissemination and providing context to the need for compliance by appearing in TV programs or at social media. I would certainly not mention all the potential options, just clarifying my point here, but it would be best to leave that last sentence of the discussion a bit more open. For example, in stead of saying that 'the government should' you can say that 'a very important prerequisite for improving compliance would be for the authorities and other stakeholders to provide......and disseminate......etc.'. That sounds less limiting but many other ways can be used to formulate this as well. 

Respons 2: Thank you for your comment. We changed the sentence in Line 359 – 363 on page 9 in the discussion to leave that last sentence of the discussion a bit more open.